# Combined Ionizing Radiation Exposure by Gamma Rays and Carbon-12 Nuclei Increases Neurotrophic Factor Content and Prevents Age-Associated Decreases in the Volume of the Sensorimotor Cortex in Rats

**DOI:** 10.3390/ijms25126725

**Published:** 2024-06-19

**Authors:** Viktor S. Kokhan, Vladimir A. Pikalov, Kirill Chaprov, Mikhail V. Gulyaev

**Affiliations:** 1V.P. Serbsky National Medical Research Centre for Psychiatry and Narcology, 119034 Moscow, Russia; 2Institute for High Energy Physics Named by A.A. Logunov of NRC “Kurchatov Institute”, 142281 Protvino, Russia; pikalov@ihep.ru; 3Institute of Physiologically Active Compounds at Federal Research Center of Problems of Chemical Physics and Medicinal Chemistry, Russian Academy of Sciences, 142432 Chernogolovka, Russia; chaprov@ipac.ac.ru; 4Faculty of Medicine, M.V. Lomonosov Moscow State University, 119991 Moscow, Russia; gulyaev@physics.msu.ru

**Keywords:** galactic cosmic rays, carbon-12 nuclei, hyperlocomotion, exploratory activity, somatosensory cortex, neurotrophins, NT-3, NT-4

## Abstract

In orbital and ground-based experiments, it has been demonstrated that ionizing radiation (IR) can stimulate the locomotor and exploratory activity of rodents, but the underlying mechanism of this phenomenon remains undisclosed. Here, we studied the effect of combined IR (0.4 Gy γ-rays and 0.14 Gy carbon-12 nuclei) on the locomotor and exploratory activity of rats, and assessed the sensorimotor cortex volume by magnetic resonance imaging-based morphometry at 1 week and 7 months post-irradiation. The sensorimotor cortex tissues were processed to determine whether the behavioral and morphologic effects were associated with changes in neurotrophin content. The irradiated rats were characterized by increased locomotor and exploratory activity, as well as novelty-seeking behavior, at 3 days post-irradiation. At the same time, only unirradiated rats experienced a significant decrease in the sensorimotor cortex volume at 7 months. While there were no significant differences at 1 week, at 7 months, the irradiated rats were characterized by higher neurotrophin-3 and neurotrophin-4 content in the sensorimotor cortex. Thus, IR prevents the age-associated decrease in the sensorimotor cortex volume, which is associated with neurotrophic and neurogenic changes. Meanwhile, IR-induced increases in locomotor activity may be the cause of the observed changes.

## 1. Introduction

The advent of the era of crewed deep-space (interplanetary) missions has significantly intensified the study of medical and biological effects of ionizing radiation (IR), whose significance increases notably when a spacecraft travels beyond the Earth’s magnetic field. Studies have shown that the effect of IR, including the most dangerous component—high-energy heavy charged particles (HZE)—is ambiguous: both negative and positive effects on central nervous system (CNS) functions have been found [1].

We have previously demonstrated that combined IR (0.4 Gy γ-rays and 0.14 Gy ^12^C, 10.4 keV/µm) results in an increase in the total distance traveled by rats in the open field test [2]. Numerous pieces of literature support IR-induced hyperlocomotion. Thus, irradiation by 1 Gy ^12^C (10.6 keV/µm) increased the locomotor activity of rats at 30, but not 60 days post-irradiation [3]. Using a more flight-relevant model of combined irradiation (H^+^, ^16^O and ^28^Si), an increase in locomotor activity was also detected at an absorbed dose of 0.5, but not 0.25 or 2 Gy [4]. Interestingly, combined IR (3 Gy γ-rays and 1.5 Gy H^+^ 0.4 keV/µm) reverses the suppression of locomotor activity induced by 30 days of antiorthostatic suspension (ground-based microgravity model) in rats [5]. Moreover, irradiation by ^56^Fe (0.5 Gy, 151.4 keV/µm) increased the locomotor activity of APP/PS1 (transgenic model of Alzheimer’s disease) female mice, as well as exploratory activity (rearing) of C57BL/6J mice in the open field test, and improved the motor coordination of APP/PS1 male mice in the rotarod test [6]. The stimulatory effect of ^56^Fe irradiation was also verified in an independent study: increased exploratory activity against novel objects placed in the open field was found in rats immediately after irradiation by ^56^Fe (1.5 Gy, 148 keV/µm) [7]. However, no change in the exploratory activity of mice was found at 2 weeks post-irradiation (0.1, 0.5, or 2 Gy) [8]. Moderate doses of H^+^ irradiation also stimulate exploratory activity of rats in different scenarios, i.e., 1 Gy H^+^ 0.4 keV/µm or 1.5 Gy H^+^ (spread-out Bragg peak) [9,10], and this effect lasts for up to 90 days post-irradiation [9]. It is noteworthy that low-dose γ-rays irradiation (63 mGy, ^60^Co) stimulates exploratory activity at 18 months post-irradiation [11]. The mechanisms responsible for the stimulatory effects of IR on locomotor and exploratory activity remain undiscovered, but there is speculation that this is part of a neuroadaptive response [1].

Neuroplasticity, as is well known, refers to the brain and neurons’ ability to adapt to new stimuli or environmental conditions. During spaceflight, neuroplasticity underlies a number of changes in the central nervous system [12,13] and can be considered as synaptic plasticity at the cellular level and changes in neuronal networks at the system level [14]. IR can cause both adaptive [11] and maladaptive [15] neuroplasticity changes, depending on dose, composition, and a number of other factors. To some extent, the possible effects of space radiation on structural changes in the brain are supported by data from astronauts: in a magnetic resonance imaging (MRI) study, the comparison of pre- and post-flight MRI scans of 27 astronauts revealed a decrease in brain gray matter volume with the exception of the somatosensory and motor cortex and, probably, the paracentral gyrus, which, in contrast, showed an increase in volume [16]. It is important to note that, during orbital flight, it is impossible to isolate the effect of IR against the background of the influence of other spaceflight factors, primarily microgravity [17,18].

A number of studies have shown that acute HZE irradiation inhibits neurogenesis at certain doses and compositions, but this blockage is transient [19,20]. At the same time, no neurogenesis inhibition was revealed after irradiation by 0.2 Gy ^28^Si (67 keV/μm) in female mice or 0.5 Gy ^56^Fe (175 keV/μm) in male mice [21,22,23]. Moreover, mice exposed to H^+^ (1 and 2 Gy, 0.24 keV/μm) are characterized by an increasing cell proliferation in the subgranular zone at 1 and 3 month post-irradiation, whereas mice exposed to ^12^C (1 Gy, 8 keV/µm) are characterized by restoration of the indices of proliferation and immature neurons in the dentate gyrus of the hippocampus at 3 months post-irradiation [19,20]. In cell-based studies, it was shown that IR (γ-rays, 661.7 keV, 4–8 Gy) increased physiological (normal) neurite growth in a dose-dependent manner [24], probably through an effect on neurotrophin expression [25].

Neurotrophins, a family of proteins that induce the survival, development, and function of neurons, are known to play a significant role in both the protection and recovery of function following CNS damage such as stroke and traumatic brain injury [26]. The expression of neurotrophins and their receptors has been shown to be extremely sensitive to X-ray irradiation. In utero X-ray irradiation (~260 keV) leads to an increase in nerve growth factor (NGF) content (1 and 2 Gy) at Postnatal Day (PND) 1, but a decrease (0.02–2 Gy) at PND21 in brain tissues. In contrast, the content of brain-derived neurotrophic factor (BDNF) was found to be increased (0.02–1 Gy) in the dentate gyrus of murine brain at PND21 [27]. Irradiation by high-dose γ-rays (10 Gy, ^60^Co) of mice results in an immediate increase in the number of neurotrophins (BDNF, neurotrophin-3 (NT-3), NGF, and GDNF) and their receptors (TrkA, TrkB, TrkC, GFRa-1, and p75NTR) in the hippocampus that persists up to 7 days after irradiation [28]. A decrease in the brain content of BDNF was found after the sequential irradiation by three nuclei (H^+^, ^16^O, and ^28^Si) at a total absorbed dose of 2 Gy (male mice only), but not the spaceflight-related 0.25 or 0.5 Gy [4]. On the contrary, the sequential irradiation by six nuclei (H^+^, ^4^He, ^16^O, ^28^Si, ^48^Ti, and ^56^Fe) at a total absorbed dose of 0.5 Gy (but not 0.25 or 2 Gy) leads to an increase in BDNF content in the neocortex [29]. A significant positive correlation between BDNF and a marker of microglial activation CD68 content was observed in both studies [4,29]. In the orbital experiment, no effect of spaceflight factors on BDNF and TrkB expression in the murine brain was detected [30].

The aim of this study was to investigate the effect of combined IR at a spaceflight-relevant dose on measures of locomotor activity, exploratory behavior, and novelty-seeking behavior, and to test the hypothesis that this effect may be associated with neuroplasticity changes in the primary motor and somatosensory cortex (sensorimotor cortex, hereinafter) of Wistar rats.

## 2. Results

### 2.1. Irradiation Increases Locomotor Activity, Anxiety, and Exploratory Behavior

The effects of irradiation (F_1,12_ = 6.5, *p* = 0.025), time (F_1,12_ = 9.5, *p* = 0.0095), and the interaction of these factors (F_1,12_ = 7.7, *p* = 0.02) reached statistical significance when the total distance traveled was analyzed. The latency time to enter the arena center was significantly influenced by irradiation (F_1,12_ = 5.2, *p* = 0.04), time (F_1,12_ = 5.2, *p* = 0.04), and the interaction of these factors was found (F_1,12_ = 5.6, *p* = 0.035). A notable effect of irradiation (F_1,12_ = 4.8, *p* = 0.048) and time (F_1,12_ = 13, *p* = 0.004) was observed when analyzing the number of rearing. At Time Point 1, irradiated R-1 rats were characterized by the higher distance traveled by 56% (*p* = 0.001), the latency time by 73% (*p* = 0.007), and the number of rearing by 65% (*p* = 0.01) compared to intact C-1 rats. At Time Point 2, the difference between groups was leveled off: the distance traveled was reduced by 45% (*p* = 0.002), the latency time by 35% (*p* = 0.007), and the number of rearing by 48% (*p* = 0.004) in R-2 rats compared to R-1 rats (Figure 1A).

A significant effect of irradiation was found when the frequency (F_1,12_ = 4.8, *p* = 0.049) and duration (F_1,12_ = 5.3, *p* = 0.04) of the exploration of an object placed in an open field was analyzed. At Time Point 1, irradiated R-1 rats showed a higher frequency and duration of object exploration, respectively, by 95% (*p* = 0.047) and 117% (*p* = 0.01) compared to intact C-1 rats. At the same time, the duration of object exploration by R-2 rats was reduced by 47% (*p* = 0.03) compared to that of R-1 rats (Figure 1B).

### 2.2. Sensorimotor Cortex Volume Is Reduced in the Intact, but Not in the Irradiated Rats through Aging

Data analysis of MRI volumetry showed a significant effect of time (F_1,12_ = 16.9, *p* = 0.0014) and interaction time × irradiation (F_1,12_ = 9.01, *p* = 0.011). We observed a 3.74% (*p* = 0.0005) decrease in sensorimotor cortex volume in intact C rats between Time Points 1 and 2 (Figure 2).

### 2.3. The Content of NT-3 and NT-4 in the Sensorimotor Cortex Increased 7 Months after the Irradiation of Rats

Statistically significant effects of irradiation (F_1,12_ = 6.7, *p* = 0.02), time (F_1,12_ = 16, *p* = 0.002), and the interaction of these factors (F_1,12_ = 27, *p* = 0.0002) were found when NT-3 content was analyzed. The statistically significant effects of irradiation (F_1,12_ = 5, *p* = 0.04) and the interaction irradiation × time factors (F_1,12_ = 5.2, *p* = 0.04) were found when NT-4 content was analyzed.

At Time Point 2, irradiated R-2 rats were characterized by higher NT-3 and NT-4 content by 40% (*p* = 0.0001) and 63% (*p* = 0.008), respectively, compared to intact C-2 rats. At the same time point, the NT-3 and NT-4 content in the R-2 rat group was greater by 43% (*p* = 0.0001) and 17% (*p* = 0.048), respectively, compared to the R-1 rat group (Figure 3).

## 3. Discussion

Combined IR exposure increases locomotor and exploratory activity, and induces novelty-seeking and anxious behavior at 3 days post-irradiation. However, this radiation-induced phenotype is transient—at 7 months post-irradiation, no differences were found between the groups of rats in the ethological analysis. It should be noted that studies involving both rodents and humans have shown a positive correlation between novelty-seeking behavior, locomotor activity, and exploratory activity [31,32]. Thus, radiation-induced changes in locomotor activity and the emotional status of rats may be mutually determined. Indeed, the radiation-induced behavioral phenotype characterized by high locomotor and exploratory activity, as well as enhanced novelty-seeking and anxiety-like behavior, is not unique and is well described in the literature [33]. A number of studies have revealed that this phenotype is associated with activation of the hypothalamic-pituitary-adrenocortical (HPA) axis [31,33]; phenotypic traits are probably controlled by the activity of catecholaminergic neural networks [32], in particular, in the locus coeruleus (LC), whose activation boosted the noradrenaline release from LC terminals [34]. These neurochemical data are fully consistent with the neurochemical profile of the brains of rats exposed to combined IR described in the previous study [2]. At the same time, pharmacological modulation of HPA axis activity provides a link between anxiety-like behavior and locomotor activity, but not exploratory activity in rats, which was irradiated at the same dose and composition [35]. Thus, activation of the HPA axis may be only part of the mechanism responsible for the radiation-induced ethological phenotype of rats.

MRI volumetry revealed a reduced volume in the sensorimotor cortex in intact rats but not in those that were irradiated. Relying on evidence that the sensorimotor cortex area undergoes thinning in naïve Wistar rats between PND80 and PND220 under standard vivarium conditions [36], we can conclude that irradiation prevents the physiologic age-associated (during the maturation period from young adult to adult in rats) thinning of this cortex region. This finding is in good agreement with data from astronauts (orbital flight), which also indicate an increase in sensorimotor cortex volume [16] and in motor cortex-cerebellar functional connectivity [37]. These findings are also supported by the data of radiation-induced disinhibition of the cortical neural network shown previously [38]. It is known that physical training and, in general, increased motor activity leads to structural changes in the brain, including the sensorimotor cortex region [39,40,41,42]. Relying on this, we hypothesized that the radiation-induced increase in locomotor activity may be responsible for the sensorimotor cortex volume increase, which counteracts its age-related thinning.

An important issue is to uncover the cellular and molecular mechanisms responsible for the preservation of sensorimotor cortex volume in irradiated rats. In line with existing literature data [1], the damage properties of IR and, in particular, HZE are undeniable. At the same time, a lesion in the neocortex induces a transient increase in the proliferation as well as neurogenesis in the subventricular zone. New neural progenitors migrate ectopically to the injured area with the assistance of blood vessels and reactive astrocytes [43,44]. Along with that, the cortical neural stem and progenitor cells, which are self-renewing and can generate neurons, astrocytes, and oligodendrocytes, may be responsible for local neuroreparation [45]. Moreover, neural stem cells are relatively radioresistant, and they can be recruited by IR-induced apoptosis [46,47]. Thus, endogenous neurogenesis and gliogenesis may be integral components of an intrinsic self-repair process under radiation-induced injury. It is important to note that the successfully synaptic integration of newborn neurons into the neocortex is necessary for the recovery process [48]. Inter alia, neurotrophins may play a significant role in facilitating this process [49,50,51].

We found that combined IR led to an increase in the content of NT-3 and NT-4 in the sensorimotor cortex. It is known that NT-3 enhances neuronal differentiation [52] and induces the development of ectopic dendrite growth [53]. Under the neocortex injuries of different genesis, NT-3 was shown to play a significant role in the regeneration of the pyramidal tract by induction of sprouting [54,55]. At the same time, studies on cell cultures of rat developing cortical neurons showed that NT-3 administration rapidly increased the frequency of spontaneous action potentials, and it synchronized excitatory synaptic activities by its reduction of inhibitory synaptic transmission mediated by GABAA receptor [56]. These data are in perfect agreement with the phenomenon of CNS disinhibition after exposure to combined IR, the mechanism of which includes the suppression of GABA-ergic inhibitory action within the neocortex [38]. Although NT-4, like BDNF, acts via the TrkB receptor, their functions differ due to differential activation of the TrkB receptor and its downstream signals [57,58]. NT-4 promotes survival as well as neurite extension and dendritic arborization in a similar manner to BDNF, via the TrkB-dependent pathway [59,60]. Interestingly, unlike BDNF, which can both decrease and increase neurotransmitter release, only the latter property has been shown for NT-4 [61]. NT-4 is also able to reduce the expression levels of pro-inflammatory cytokines, improve neurological function, and attenuate neuroinflammation [62]. At the same time, anti-inflammatory therapy can prevent the age-related loss of brain tissue volume [63]. It is important to note that all neurotrophins are involved in “rewiring” the neocortex connectome [53,64]. Finally, the observed increase in neurotrophin content aligns well with studies on the effects of IR on the neurogenic microenvironment. Indeed, using cell transplantation technology, it has been shown that IR does not reduce the potential for further neurogenesis but, on the contrary, significantly enhances the engraftment potential of transplanted neural and multipotent astrocytic stem cells, which indicates the relative intactness of the microenvironment [65,66].

Our study has some limitations. Despite the use of a spaceflight-relevant total equivalent dose, the absence of moderate- and high-LET heavy nuclei makes this model imperfect. The results obtained need to be validated using better models that assume multicomponent low-dose chronic or fractionated exposures that reproduce the natural radiation environment in outer space. In the test of exploratory activity assessment, the object’s appeal to the rat is crucial. We used netsuke, which showed high attractiveness to rats in our previous studies. However, using different objects may significantly affect the results.

## 4. Materials and Methods

### 4.1. Animals

Twenty-eight male 3-month-old Wistar rats weighing 270–290 g were used. Animals were maintained in groups of three or four per cage (50 × 36 × 20 cm; length × width × height) in a standard environment (12-h light/dark cycle, 19–22 °C and 50–60% relative humidity) with food and water ad libitum.

### 4.2. Study Timeline

The timeline of the study is shown in Figure 4. Five days before the experiment, all rats were weighed and distributed into four groups (n = 7 for all groups) so that animals with the same weight were in different groups according to the minimization approach in randomization [67]. At 99 days of age, 2 groups of rats were irradiated (R rats), and the other 2 groups remained intact and served as controls (C rats). One group of C rats and 1 group of R rats were subjected to ethological and MRI analysis twice: within 1 week (C-1 and R-1 data; Time Point 1) and 7 months (C-2 and R-2 data; Time Point 2) post-irradiation; at 322 days of age, rats from these groups were euthanized for biochemical analysis (data for Time Point 2). The remaining groups of rats (C and R) were euthanized at Day 7 post-irradiation for biochemical analysis (data for Time Point 1).

### 4.3. Irradiation Procedures and Dosimetry

Irradiation by 661.7 keV γ-rays was performed with a GOBO-60 containing a ^137^Cs source with certified activity of a 72 g equivalent of ^226^Ra. Mice were whole-body irradiated daily (~16.7 mGy/h) at a total absorbed dose of 400 ± 30 mGy. The absorbed dose was measured with thermoluminescent monocrystalline DTG-4 (LiF-Mg, Ti) detectors (A.P. Vinogradov Institute of Geochemistry SB RAS, Russia). Detector annealing and dose calculation were carried out on the Harshaw TLD Model 3500 manual readers (Thermo Fisher Scientific, Waltham, MA, USA).

Twenty-four hours after γ-rays irradiation, the head of the experimental animals was irradiated by ^12^C nuclei (450 MeV/u; linear energy transfer (LET): 10.4 keV/μm) at a total absorbed dose of 140 ± 10 mGy in the U-70 accelerator (NRC “Kurchatov Institute”—IHEP, Protvino, Russia): A rat was placed in a special poly(methyl methacrylate) case that restricted movement and was positioned so that only the head and a small area of the neck fell into the ^12^C^6+^ ion beam, which was formed by a carbon collimator (density: 1.8 g/cm^3^; thickness: 50 cm; diameter of beam: 65 mm). Dosimetric monitoring of ion irradiation was performed using a dosimeter DKS-AT5350/1 (Atomtex, Minsk, Belarus) with a TM30010-1 ionization chamber (PTW, Freiburg, Germany).

The control group was housed in the same room as the irradiated rats, but behind the GOBO-60 source, thus not exposed to the γ-rays flux. After that, the control group of rats was transported together with irradiated animals and placed in the cases, but was kept separately in another laboratory room and thus not exposed to a ^12^C^6+^ ion beam.

We used the combined irradiation model relying on the following arguments. Despite the small fraction of γ-quanta in galactic cosmic rays (GCR), they represent a significant fraction of secondary radiation. However, the main purpose of using γ-rays pre-irradiation with a relative low dose rate was to provide tissue sensitization. Indeed, during space flight, before living tissue is hit by HZE, it will experience multiple penetrations by bremsstrahlung X-rays, δ-rays, and protons. Studies have shown that such low-dose low-LET pre-irradiation significantly modulates the organism’s response to irradiation by HZE [68]. When choosing the HZE component, we relied on the fact that ^12^C^6+^ is the most common nucleus after H^+^ and ^4^He^2+^ in the composition of GCR [69], while the LET value was near the median value on the GCR LET spectrum measured on the Mars surface and during the Earth–Mars cruise [1]. Based on the dosimetric study, low-LET exposure will be predominant during spaceflight [70]. Moreover, during the realization of crewed deep-space missions, more advanced shielding will be used, which will also shift the particle spectrum towards the low-LET component and may reduce the equivalent dose by up to 30% [71,72]. The total equivalent dose in the used irradiation model (~0.8 Sv, for brain tissue, based on [73]) is relevant to that of a hypothetical 860-day Martian mission [70]. It should be noted that the used model of combined irradiation will allow us to make a correct comparison of the obtained results with our previous studies [2,35,38].

### 4.4. Open Field Test with Object Exploration

The test was performed according to the method described by Pecaut et al. [8] with some modifications. A cylindrical box (d = 100 cm and the wall h = 50 cm) for the open field test was used. The floor (arena) of the box was with holes (d = 2 cm), which imitate minks. The arena was evenly illuminated (60 lx). The rats were placed in the arena close to a wall, and the rates of horizontal and vertical activity were registered for 5 min by RealTimer v.1.30 software (OpenScience, Moscow, Russia). After 5 min, the response to novelty was evaluated in the same arena by placing an object (by hand, using laboratory gloves to minimize identifying odors) near the wall on the opposite side from the rat. The frequency of approaches (paw and/or muzzle touches; 3 cm zone around the object—the hysteresis zone) and the duration of contact (paw and/or muzzle touches) with the object were recorded for 3 min. After each animal testing, the object and the arena were wiped with 70% ethanol and dried with a hair dryer. Two netsuke that are very different in shape but not in size—God Hotei and God Ganesha—were used as objects, respectively, at Time Points 1 and 2.

### 4.5. Magnetic Resonance Imaging-Based Morphometry

MRI studies were performed on a 7.05 T MR scanner Bruker BioSpec 70/30 USR driven by ParaVision v.5.1 software (Bruker, Ettlingen, Germany). The animals were anesthetized in a specialized plastic container with isoflurane at a concentration of 4.5% at an oxygen flow of 1 L/min, and then adjusted to 1.5% during scanning. Gas anesthesia was administered using an isoflurane vaporizer from Ugo Basile S.R.L. (Comerio, Italy) and the oxygen concentrator JAY-10 (Longfian Sitech, Baoding, China). Special platinum sticks were inserted into the ears for additional immobilization of the animal’s head. Axial, coronal, and sagittal T2-weighted images were obtained using a spin-echo pulse sequence RARE (rapid acquisition with relaxation enhancement) with the following scan parameters: repetition time (TR): 5500 ms; effective echo time (TE_eff_): 47.5 ms; number of echoes collected during each repetition (RARE-factor): 6; field of view: 3.12 × 2.34 cm^2^; matrix: 208 × 156; spatial resolution: 0.15 × 0.15 mm^2^; slice thickness: 0.5 mm; no gaps; number of averages: 4; number of slices: 30, 20, and 29, respectively, for the axial, coronal, and sagittal views. The processing of the MR images was carried out in a freely distributed program ImageJ v.1.51j8 [74].

The cortical area whose volume was precisely estimated (Figure 5) included the primary motor cortex (M1) and the primary somatosensory cortex (three zones: S1FL, S1HL, and dysgranular S1DZ; the sensorimotor cortex). To identify the target area of the neocortex, we employed data from both the in vivo rat atlas by Schwarz et al. [75] and the rat brain atlas by Paxinos and Watson [76].

### 4.6. Tissue Collection

Rats from the experimental and control groups were euthanized by decapitation. The sensorimotor cortex was dissected in a non-precision manner on thermoelectric cooling surface (+2 °C) and immediately frozen in liquid nitrogen until analyzed.

### 4.7. Multiplex Assay

Tissue samples were mechanically homogenized in polypropylene tubes on ice using a polytetrafluoroethylene pestle (600 rpm, ~20 s) in a lysis buffer (+20 °C; 1:20 *v*/*v*) consisting of the following components: 20 mmol/L Tris-HCl (pH 7.5), 150 mmol/L NaCl, 0.05% *v*/*v* Tween-20, and 1% *v*/*v* protease inhibitor cocktail II (ab201116, Abcam, Cambridge, UK). The obtained homogenates were centrifuged at 12,000× *g* for 15 min at 3 °C, and the supernatant was taken for further analysis. Protein content in samples was determined by the Bradford method using a Quick Start Bradford Protein Assay kit (Bio-Rad, Hercules, CA, USA). For analysis, samples were diluted by a lysis buffer to a target protein concentration in the range of 0.6–1 mg/mL, if necessary.

A multiplex assay was performed using the commercially available IS-135-Rat kit (Cloud-Clone Corp., Wuhan, China) to determine NGF, BDNF, NT-3, and NT-4 concentration. Bead preparation, handling, and plate processing were conducted according to the manufacturer’s protocol. Plates were washed using a Bio-Plex Pro Wash Station (Bio-Rad, Hercules, CA, USA) and read using a Bio-Plex MAGPIX Multiplex Reader (Bio-Rad, Hercules, CA, USA). The concentration of neurotrophins in the tested samples was determined automatically with standard calibration dilutions using Bio-Plex Manager Software v.6.1 (equipment management and initial data processing) and Bio-Plex Data Pro Software v.1.2 (Bio-Rad, Hercules, CA, USA) for final data processing. The content of target proteins was normalized to total protein in the sample.

### 4.8. Data Processing

Data were presented as the mean ± standard deviation (SD). Data processing was performed using Statistica 12 software (StatSoft Inc., Tulsa, OK, USA). The Shapiro–Wilk test was used to assess the normality of the data distribution; when *p* > 0.05, parametric analysis methods were used. Thus, data were processed using repeated measures ANOVA. Duncan’s post hoc test was conducted if necessary.

## Figures and Tables

**Figure 1 ijms-25-06725-f001:**
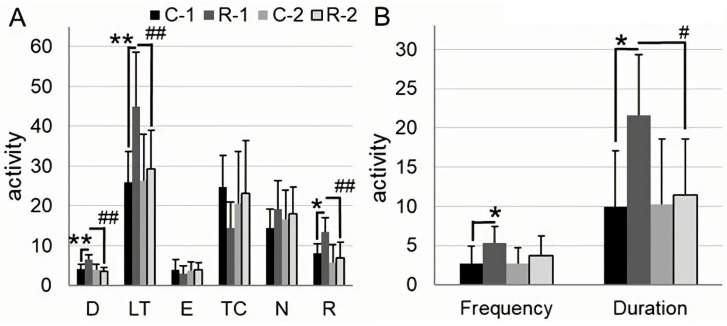
Open field test. (**A**) open field test activity for 5 min; D: distance traveled, m; LT: time of first entry to the arena center (latency time), s; E: center arena entries; TC: time spent at arena center, s; N: number of hole-poking; R: number of rearing. (**B**) object exploration for 3 min: Frequency: number of approaches (paw and/or muzzle touches; 3 cm zone around the object—the hysteresis zone) of the rat to the placed object; duration: time of contact (paw and/or muzzle touches) of the rat with the placed object, s. Bar charts show mean + SD. R: irradiated rats, n = 7; C: control intact rats, n = 7; numeric indices indicate the time points of the analysis: 1 or 2. Asterisk indicates statistically significant differences between the groups within one time point of analysis (*: *p* < 0.05, **: *p* < 0.01; post hoc Duncan’s test). Hash indicates intragroup statistically significant differences between the 1st and 2nd time point of analysis (#: *p* < 0.05, ##: *p* < 0.01; post hoc Duncan’s test).

**Figure 2 ijms-25-06725-f002:**
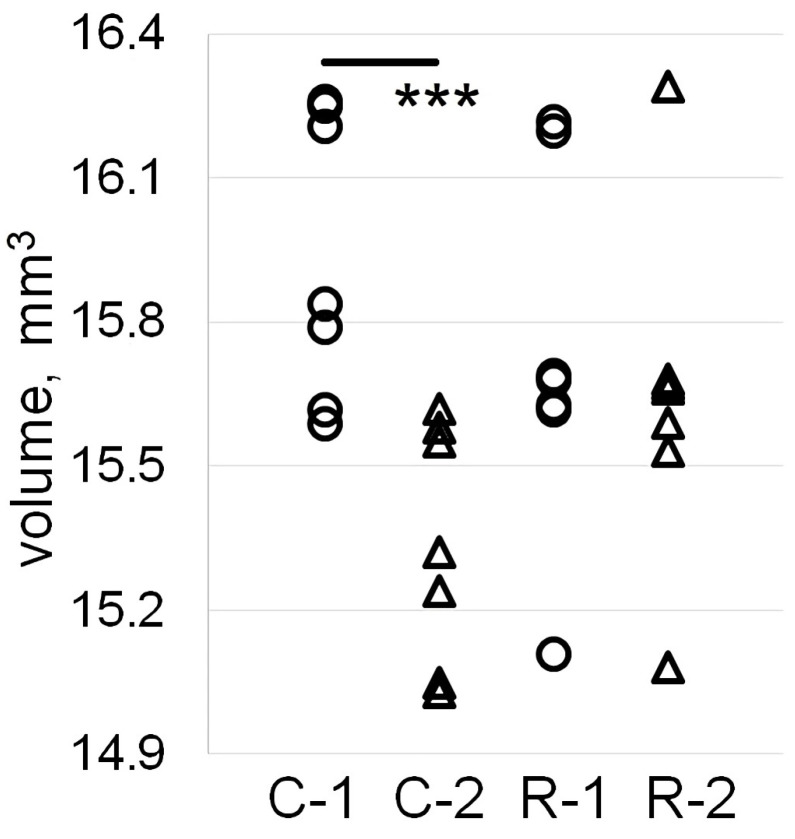
Magnetic resonance imaging volumetry of the sensorimotor cortex. Individual cortical volume data are presented as circles for Time Point 1 and triangles for Time Point 2; n = 7 for C (intact) and R (irradiated) groups of rats. Asterisk indicates intragroup statistically significant differences between Time Points 1 and 2 (***: *p* < 0.001; post hoc Duncan’s test).

**Figure 3 ijms-25-06725-f003:**
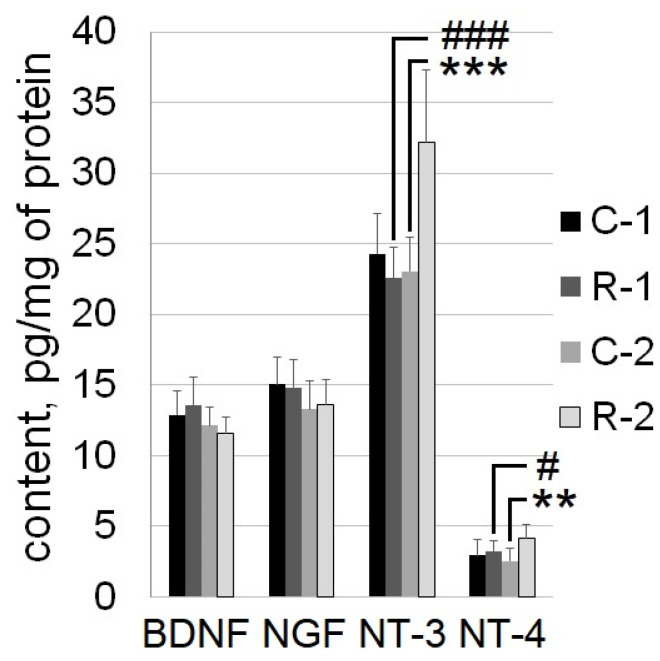
Content of neurotrophins (BDNF, NGF, NT-3, and NT-4) in the rats’ sensorimotor cortex. Bar charts show mean + SD. R: irradiated rats, n = 7; C: control intact rats, n = 7; numeric indices indicate the time points of the analysis: 1 or 2. Asterisk indicates statistically significant differences between the groups within a single time point of analysis (**: *p* < 0.01, ***: *p* < 0.001; post hoc Duncan’s test). Hash indicates intragroup statistically significant differences between Time Points 1 and 2 (#: *p* < 0.05, ###: *p* < 0.001; post hoc Duncan’s test).

**Figure 4 ijms-25-06725-f004:**
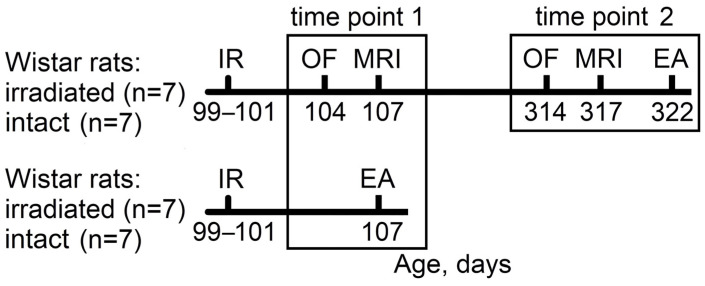
Study timeline. IR: exposure to combined ionizing radiation: 400 mGy γ-rays (~16.7 mGy/h) and, 24 h later, 0.14 Gy ^12^C nuclei, 450 MeV/u; OF: open field test; MRI: magnetic resonance imaging-based brain morphometry; EA: euthanasia. The ages of the rats are shown in days. Two time points in the analysis were used to assess the dynamics of changes: 1: within one week post-irradiation; 2: seven months post-irradiation.

**Figure 5 ijms-25-06725-f005:**
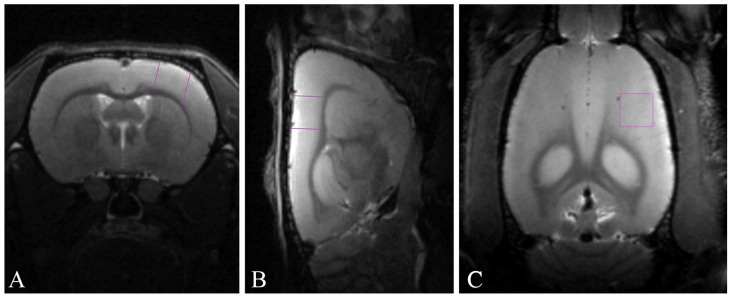
Representative T2-weighted MR images of rat brain with analysis area highlighted by purple lines: (**A**) axial projections; (**B**) sagittal projections; (**C**) coronal projections.

## Data Availability

The data presented in this study are openly available in Mendeley Data, link: Kokhan, Viktor (2024), “Combined Ionizing Radiation Exposure by Gamma Rays and carbon-12 Nuclei Increases Neurotrophic Factor Content and Prevents Age-Associated Decreases in the Volume of the Sensorimotor Cortex in Rats”, Mendeley Data, V1, DOI: 10.17632/nwj854ppyd.1; https://data.mendeley.com/datasets/nwj854ppyd/1 (accessed date: 15 June 2024).

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
