# Peer review of "Combined Ionizing Radiation Exposure by Gamma Rays and Carbon-12 Nuclei Increases Neurotrophic Factor Content and Prevents Age-Associated Decreases in the Volume of the Sensorimotor Cortex in Rats"

_ijms, 2024, doi:10.3390/ijms25126725_

Round 1

Reviewer 1 Report

Comments and Suggestions for Authors

Dear authors,

please find my comments in the attached file.

Best regards

Comments on the Quality of English Language

The English can be improved in some sentences. Suggestions are given in the file.

Reviewer 2 Report

Comments and Suggestions for Authors

I assess this manuscript high. It is an interesting subject and very challenging. I have some comments/questions:

1. The introduction is too long and feels redundant. Please make it a little more concise and easy for readers to understand.

2. What is "F1,12" described in the results section? Also, please unify their notation.

3. Why did we set the time points of 1 week and 3 months after irradiation?

4.  As the title states, radiation exposure prevents the age-associated decrease in the sensorimotor cortex volume of rats. However, the lifespan of rats is 3 to 4 years, so analysis over a longer time span is required to draw such a conclusion.

5. Please write the unit for the vertical axis in Fig.1.

6. Please post representative MRI images taken of rats under each condition. The amount of motor cortex varies among radiation-exposed groups, and I think we need to increase the number of N a little more.

Author Response

We thank the reviewer for the comments on the manuscript content.

Comments and Suggestions for Authors

I assess this manuscript high. It is an interesting subject and very challenging. I have some comments/questions:

  1. The introduction is too long and feels redundant. Please make it a little more concise and easy for readers to understand.

Answer:

We carefully analyzed the “Introduction” section and found it possible to reduce the amount of text by removing the characteristics of radiation. However, in this case it would be difficult for the reader to compare the data correctly, since biological effects of ionizing radiation depend on a set of characteristics (dose, electromagnetic or corpuscular nature, nucleus size, linear energy transfer). Therefore, we kindly ask to leave the section unchanged. It should be noted, the comments of other reviewers allowed us to improve it and make it more understandable to the reader.

  1. What is "F1,12" described in the results section? Also, please unify their notation.

Answer: F is the absolute value of Fisher's criterion, the digits in the subscript indicate the number of degrees of freedom. In the corrected version of the manuscript, all degrees of freedom were formatted with a subscript.

  1. Why did we set the time points of 1 week and 3 months after irradiation?

Answer:

We used 2 analysis points - within 1 week (first point) and 7 months (second point) after irradiation. The first point was chosen relying on literature data that suggest that radiation-induced novelty-seeking behavior is transient and undetectable as early as 2 weeks after irradiation (doi: 10.1016/j.asr.2003.12.011, doi: 10.1667/rrr3205). We present these data in the “Introduction” Line 52-55. The second time point was chosen based on several considerations. Firstly, we wanted to compare the obtained data with earlier data on the disinhibitory effects of irradiation within a neocortex, which were also obtained 7 months after irradiation (doi: 10.1016/j.neuroscience.2019.08.009). Secondly, we had to choose a period between the rat age from 220 to 365 days, when, respectively, there is a physiological decrease in sensorimotor cortex volume in Wistar rats appear (doi: 10.1111/adb.12364) and when the effect of irradiation can be leveled off (doi: 10.1016/j.ijrobp.2008.02.015, doi: 10.1007/s00411-009-0220-5) and we will not be able to detect a radiation-induced changes.

  1. As the title states, radiation exposure prevents the age-associated decrease in the sensorimotor cortex volume of rats. However, the lifespan of rats is 3 to 4 years, so analysis over a longer time span is required to draw such a conclusion.

Answer:

We intended to focus on the physiological age-associated (between age 107 and 322 postnatal days) thinning of the sensorimotor cortex, rather than aging-associated.

To avoid any misunderstanding, we clarified that it is “age-associated (during the maturation period from young adult to adult in rats)” in the revised version of the manuscript (Line 198).

  1. Please write the unit for the vertical axis in Fig.1.

Answer:

Each behavioral parameter in Fig. 1 has a different unit of measurement: time (s), distance (m), and number of acts (number of rearing, number of exits to the center, etc.). We grouped them under the overarching term “activity” and gave an explanation in the description to the figure (Line 132-141). While we prefer to keep the figure as it is, if the reviewer insists, we are willing to split the figure into  separate ones, each with the appropriate units indicated on the y-axis. However, we believe that such a division would unnecessarily increase the volume of the manuscript.

  1. Please post representative MRI images taken of rats under each condition. The amount of motor cortex varies among radiation-exposed groups, and I think we need to increase the number of N a little more.

Answer:

Although we could provide MRT images for each group, the minor differences (up to 7% for the control group and up to 3% for the irradiated group) make them almost indistinguishable to the naked eye, potentially leading the reader to mistakenly assume they are identical. Thus, these images lack representativeness. We present the primary data of cortical volume measurements in the study dataset (doi: 10.17632/nwj854ppyd.1). We kindly request to retain only the image indicating the MR-morphometry area.

Indeed, the sample size is an extremely important parameter for the statistical power of a study. The presented work is a pilot study with a small but sufficient for statistical analysis group size. Since the results of the study confirmed our hypothesis, we plan to conduct a future study with a larger group size and with extended biochemical and molecular analyses.

Reviewer 3 Report

Comments and Suggestions for Authors

This is a well designed study that aims at elucidation of the possible mechanisms responsible for neuroplasticity related to irradiation with low doses of gamma and particulate radiation -  environmental factors that are relevant for deep space exploration. The authors built a convincing argument with both behavioural, imaging and biochemical data showing a correlational relationship between long standing functional and structural CNS changes and radiation-induced changes in neurotrophin levels in selected regions of rats' brain. Subsequently they built an interesting argument that relatively low dose mixted radiation exposure might prevent physiological, age-related changes in the cerebral cortex. Being a worthy extension of their previously published studies on the same topic I found the paper at hand worth publishing in IJMS in a present form.

Author Response

We are extremely grateful for such an appreciation of our study.